# Enhancing Cancer Diagnosis with Real-Time Feedback: Tumor Metabolism through Hyperpolarized 1-^13^C Pyruvate MRSI

**DOI:** 10.3390/metabo13050606

**Published:** 2023-04-28

**Authors:** Gaurav Sharma, José S. Enriquez, Ryan Armijo, Muxin Wang, Pratip Bhattacharya, Shivanand Pudakalakatti

**Affiliations:** 1Department of Cardiovascular & Thoracic Surgery, UT Southwestern Medical Center, Dallas, TX 75390, USA; gaurav.sharma@utsouthwestern.edu; 2Advanced Imaging Research Center, University of Texas Southwestern Medical Center, Dallas, TX 75390, USA; 3Department of Cancer Systems Imaging, The University of Texas MD Anderson Cancer Center, Houston, TX 75390, USA; jsenriquez@mdanderson.org (J.S.E.); rcarmijo@mdanderson.org (R.A.); mwang15@mdanderson.org (M.W.); pkbhattacharya@mdanderson.org (P.B.); 4Graduate School of Biomedical Sciences, The University of Texas MD Anderson Cancer Center, Houston, TX 75390, USA

**Keywords:** hyperpolarized (HP) ^13^C magnetic resonance spectroscopic imaging (MRSI), cancer metabolism, pyruvate-to-lactate metabolic reprogramming, therapeutic intervention, precision medicine, biomarkers, artificial intelligence

## Abstract

This review article discusses the potential of hyperpolarized (HP) ^13^C magnetic resonance spectroscopic imaging (MRSI) as a noninvasive technique for identifying altered metabolism in various cancer types. Hyperpolarization significantly improves the signal-to-noise ratio for the identification of ^13^C-labeled metabolites, enabling dynamic and real-time imaging of the conversion of [1-^13^C] pyruvate to [1-^13^C] lactate and/or [1-^13^C] alanine. The technique has shown promise in identifying upregulated glycolysis in most cancers, as compared to normal cells, and detecting successful treatment responses at an earlier stage than multiparametric MRI in breast and prostate cancer patients. The review provides a concise overview of the applications of HP [1-^13^C] pyruvate MRSI in various cancer systems, highlighting its potential for use in preclinical and clinical investigations, precision medicine, and long-term studies of therapeutic response. The article also discusses emerging frontiers in the field, such as combining multiple metabolic imaging techniques with HP MRSI for a more comprehensive view of cancer metabolism, and leveraging artificial intelligence to develop real-time, actionable biomarkers for early detection, assessing aggressiveness, and interrogating the early efficacy of therapies.

## 1. Introduction 

Metabolic reprogramming, a characteristic feature of cancer, encompasses the Warburg effect that is apparent in various cancer types [1,2]. Noninvasive imaging techniques that can read out molecular processes related to tumor development and proliferation are, therefore, necessary [3]. Tumor bioenergetics and proliferation are intricately tied to metabolic reprogramming, making it feasible to develop noninvasive metabolic imaging methods that can assess tumor burden and response to treatment. Currently, positron emission tomography (PET) is the conventional approach for clinical cancer imaging as it leverages these principles [4]. However, ionizing radiation considerably increases the incidence of both cancer and heritable effects, and modalities involving ionizing radiation pose a challenge to the long-term evaluation of treatment monitoring in these cancer patients [5]. An innovative approach to identify altered metabolism is carbon-13 magnetic resonance spectroscopic imaging (MRSI), which involves injecting hyperpolarized (HP) [1-^13^C] pyruvate [3]. Over the past decade, this technique has shown promise in identifying upregulated glycolysis in most cancers, as compared to normal cells [6]. The method enables dynamic, noninvasive imaging of the biochemical conversion of [1-^13^C] pyruvate to [1-^13^C] lactate and/or [1-^13^C] alanine (Figure 1). Hyperpolarization significantly improves the signal-to-noise ratio for the identification of ^13^C-labeled metabolites [7].

Hyperpolarized ^13^C metabolic imaging is now being investigated for imaging of several human diseases and has shown encouraging preclinical findings in the areas of cancer and cardiovascular diseases [8]. In patients with prostate [9,10,11], brain [12], renal [13], breast [14,15], and pancreatic [16] cancer, the first translational studies showed the technique to be safe and practicable. Some of these studies have also shown that the technique could detect successful treatment responses in patients with breast and prostate cancer at an earlier stage before multiparametric MRI captures the morphological changes. The use of metabolic imaging, such as HP ^13^C MRSI, for accurate, individualized imaging of cancer patients offers considerable potential. In an era of precision medicine, the detection and treatment of tumors are progressing toward a new approach. This approach involves examining the genetic makeup of each patient prior to treatment, which enables the administration of personalized drugs tailored to specific mutation patterns. Identifying suitable biomarkers or companion diagnostics that can provide an early indication of treatment response is a vital component of precision medicine [17,18]. HP ^13^C metabolic imaging has the potential to offer a noninvasive readout of underlying tumor genomes and support tumor diagnosis and/or prognosis since oncogenic events also promote tumor metabolic reprogramming [6]. While advanced multiparametric MRI is increasingly being used to monitor cancer patients undergoing treatment, there is a need to develop more precise, sensitive, and real-time biomarkers for treatment response at the early stages of the disease. Additionally, HP ^13^C MRSI has the capacity to offer an early readout of therapeutic response once the patient has received care using precision medicine techniques. This approach is crucial for long-term studies of therapeutic response, particularly for tumors that are anatomically inaccessible. The first hyperpolarized ^13^C agent to successfully enter the clinic was [1-^13^C] pyruvate, in keeping with its frequent usage in preclinical investigations. Clinical studies with HP [1-^13^C] pyruvate are now being conducted on various tumor types all around the world. This article offers a succinct summary of the potential applications of HP [1-^13^C] pyruvate magnetic resonance spectroscopic imaging (MRSI) across different cancer systems. It sheds light on the potential of this technique utilized in preclinical and clinical investigations, as well as in the field of precision medicine. Furthermore, it emphasizes the significance of incorporating HP [1-^13^C] pyruvate MRSI in long-term studies of therapeutic response, as it can provide valuable insights into the effectiveness of cancer treatments over time. The ability of HP [1-^13^C] pyruvate MRSI to noninvasively visualize metabolic changes in cancer cells offers a promising avenue for the development of novel cancer diagnostic and treatment strategies. Thus, it has the potential to revolutionize cancer management in the future.

## 2. Brain Cancer 

Brain tumors are highly resistant to standard and innovative treatments due to the heterogeneity, hypoxia, and blood–brain barrier [19]. This makes them one of the deadliest forms of cancer [17]. In preclinical settings, HP [1-^13^C] pyruvate was used to interrogate brain tumor metabolic pathways and to assess treatment response in cancer models. When comparing rats with human glioblastoma (GBM) xenografts to normal rats, elevated lactate-to-pyruvate ratio was observed in GBM rats, demonstrating the ability of HP MRSI to differentiate between normal brain tissue and cancerous tissue [20]. Lower-grade gliomas, which most commonly have the isocitrate dehydrogenase 1 (IDH1) mutation, were investigated using hyperpolarized [1-^13^C] pyruvate MRSI and presented with a lower pyruvate-to-lactate conversion compared to GBM models. This is consistent with previous reports of mutant IDH1 gliomas being associated with low levels of lactate dehydrogenase A [21]. In the investigation of other GBM metabolic pathways, HP [1-^13^C] pyruvate was employed in the detection of pyruvate dehydrogenase flux modulated by dichloroacetate, providing a valuable tool for the understanding of GBM metabolism [22]. In another study, hyperpolarized [1-^13^C] pyruvate was used to measure radiotherapy treatment response, which was validated in a glioma tumor model where FDG-PET detection is limited by high background levels from normal tissue [23]. Importantly, HP [1-^13^C] pyruvate was able to detect a decreased pyruvate-to-lactate conversion before inhibition of tumor growth post-treatment occurred, allowing for earlier feedback on treatment efficacy [24]. Another area of clinical brain cancer imaging where the application of HP [1-^13^C] pyruvate can make an impact in determination of pseudo-progression (an increase in tumor size might be due to infiltration of immune cells and accumulation of fluids, followed by a regression of tumor size) following different brain cancer therapies. This has been successfully shown in preclinical models with hyperpolarized metabolic imaging after radiation [25].

HP [1-^13^C] pyruvate MRSI has shown the potential to distinguish brain tumor in eight human subjects from normal healthy brain according to metabolism (four high-grade primary GBM, three low-grade oligodendrogliomas, and one low-grade astrocytoma) [12]. Lactate was found in tumor areas, whereas bicarbonate and lactate were found in normal contralateral brain, indicating that HP [1-^13^C] pyruvate was delivered effectively across the blood–brain barrier. The decrease in bicarbonate-to-lactate ratio in the lesion compared to normal-appearing brain (NAB) tissue shows the potential to distinguish between tumor and normal brain. Another study, which included subjects with metastatic brain cancer, confirmed these findings [26]. In individuals with primary brain tumors, lactate generation from HP [1-^13^C] pyruvate was comparable between tumor and contralateral brain. Interestingly, the observation of substantial pyruvate-to-lactate conversion in normal human brain in these early clinical investigations contrasts with the preclinical evidence available [27,28], which indicated relatively little lactate generation in normal rodent brain. These results suggest that the use of HP [1-^13^C] pyruvate may be a valuable tool for investigating brain tumor metabolism and could potentially lead to more effective treatment options for patients with brain tumors.

## 3. Pancreatic Cancer 

Another heavily studied cancer system with HP [1-^13^C] pyruvate is pancreatic cancer, specifically pancreatic ductal adenocarcinoma (PDAC). PDAC is one of the deadliest cancers, often diagnosed at later stages due to its asymptomatic early presentations [29]. Early detection is crucial for effective treatment, and new imaging biomarkers are required to accomplish this.

There is evidence of glycolytic metabolic alterations in pre-malignant stages, which can be exploited by HP [1-^13^C] pyruvate MRSI [30]. It was reported that the aggressiveness of PDAC is directly correlated to pyruvate-to-lactate conversion measured using HP [1-^13^C] pyruvate MRSI in a patient-derived xenograft (PDX) mouse model [31]. In the more aggressive tumors, the pyruvate-to-lactate conversion was higher (Figure 2). Another study was conducted on detecting PDAC in early stages in mice using HP [1-^13^C] pyruvate. The study involved KPC (K-Ras and p53 mutations) and KC (K-Ras mutation) mice, genetically engineered to develop PanIN (pancreatic intraepithelial neoplasia). The mice were imaged at different stages of development from the precursor lesions PanIN to PDAC, and the fluxes from HP [1-^13^C] pyruvate to lactate and alanine were measured. The alanine/lactate ratio was discovered as an imaging biomarker that decreased with disease progression from normal tissue to low-grade PanIN to high-grade PanIN and finally to PDAC [32]. These results were similar to the results on PDX models in association with aggressiveness unlike genetically engineered models. These similar results from different researchers demonstrate the robustness of the real-time conversion kinetic rate constants (*k_PA_* and *k_PL_*) as the biomarkers for the detection of premalignant pancreatic lesions [33]. *k_PA_* and *k_PL_* are the kinetic rate constants for pyruvate-to-alanine and pyruvate-to-lactate biochemical reactions.

The biochemical conversion of HP [1-^13^C] pyruvate to lactate can be used to determine the treatment efficacy earlier than the appearance of morphological changes. One of the studies tested a hypoxia-activated prodrug (TH-302) as a monotherapy and in combination with pyruvate (not the HP probe) on three subcutaneous (Hs766t, MIAPaCa-2, and SU.86.86 cells) PDX PDAC models derived from different patients [34]. Metabolic phenotypes of these models were evaluated using HP [1-^13^C] pyruvate, and Hs766t and MIAPaCa-2 showed more glycolytic expression. However, SU.86.86 was resistant to the TH302 hypoxic prodrug because it was less hypoxic, with oxygen levels of 17.6 ±  2.6 mmHg. The mice were treated five times a week for 2 weeks, and tumor sizes were measured to determine the treatment efficacy. Hs766t and MIAPaCa-2 showed good responses to TH302 compared to SU.86.86 [34]. The treatment efficacy of LDHA inhibitor (drug FX11) on 15 patient-derived PDAC mouse models was tested. LDHA is an enzyme that converts pyruvate to lactate in the presence of cofactor NADH and is upregulated in pancreatic cancer. Inhibition of this enzyme could induce the metabolic vulnerability of cancer and can be utilized as a potential treatment. The treatment efficacy was tested with HP [1-^13^C] pyruvate MRSI experiments, and the experiments were conducted before treatment and 7 days after treatment. The mice responding to treatment with FX11 showed a decreased lactate-to-pyruvate ratio, whereas non-responders showed an increased lactate-to-pyruvate ratio after 7 days of treatment [35].

Another system that has been targeted is the quinone oxidoreductase 1 (NQO1)-mediated redox cycle, which can be targeted by the β-lapachone chemotherapeutic drug. This drug results in elevated production of superoxide and peroxide while depleting nicotinamide adenine dinucleotide (NAD^+^) as a consequence of DNA damage and hyperactivation of poly (ADP-ribose) polymerase. The decreased concentration of NAD^+^ has impact on pyruvate-to-lactate conversion; thus, the drug efficacy can be tested with HP [1-^13^C] pyruvate MRSI. The β-lapachone drug was tested on patient-derived MIAPaCa2 cells (which were NQO1^+^, and sensitive to β-lapachone) in vitro. This study demonstrated a decrease in glycolytic flux after treatment, making use of HP [1-^13^C] pyruvate a promising technique to determine the treatment efficacy of β-lapachone in patients with PDAC tumors and upregulated NQO1 [30,36]. Response to radiation therapy has also been predicted to be indirectly measured by HP [1-^13^C] pyruvate. Radiation therapy generally generates reactive oxygen species (ROS), and this oxidative stress and oxidative damage can be measured by HP MRSI. This can be exploited by the fact that pyruvate-to-lactate conversion requires reducing equivalents, which demonstrates why HP pyruvate-to-lactate conversion can be employed to predict the response to radiation therapy in solid tumors in animal models [37]. Pancreatic cancer studies with HP [1-^13^C] pyruvate have also been conducted in a clinical setting. In a pilot study, more than 30 s after the injection, it was demonstrated that hyperpolarized [1-^13^C] pyruvate MRSI effectively distinguished pancreatic tumor tissue from surrounding tissue by producing [1-^13^C] lactate and [1-^13^C] alanine, with no harmful side-effects observed after injection of pyruvate [16].

## 4. Ovarian Cancer

Among cancer-related deaths in women, ovarian cancer ranks fifth. Survival rates of over 95% for more than 5 years are possible if ovarian cancer is detected early. Unfortunately, reliable biomarkers for the early detection of ovarian cancer are lacking. Transvaginal ultrasound and blood CA-125 levels often give false positives due to functional cysts or a variety of other medical conditions [38]. The need for early detection methods is imperative for women at a high risk of developing ovarian cancer, such as women who harbor *BRCA* germline mutations. The current resolution of clinical standard-of-care imaging cannot effectively detect tumors less than 1 cm. This underpins the need for developing high-resolution imaging and for exploring hyperpolarized metabolic imaging and tumor-specific biomarkers to find smaller tumors. 

Recently, HP [1-^13^C] pyruvate has been explored to diagnose and analyze therapeutic responses in ovarian cancer in a preclinical setting. The nude SKOV3 ovarian cancer mouse model was treated with the multityrosine kinase inhibitor pazopanib, and the treatment efficacy was evaluated using hyperpolarized [1-^13^C] pyruvate and ^18^F-FDG-PET. No statistically significant difference between pazopanib-pretreated mice and vehicle control mice was observed by HP [1-^13^C] pyruvate MRSI and ^18^F-FDG-PET. After treatment, ^18^F-FDG-PET and anatomical MRI failed to determine the efficacy of pazopanib treatment, whereas HP [1-^13^C] pyruvate MRSI showed significant differences in the lactate-to-pyruvate ratios between pazopanib pretreated and post-treated animal models, as well compared to vehicle control cohorts [39].

## 5. Prostate Cancer 

Prostate cancer (PCa) is the second most common reason for cancer-caused fatalities among men [40,41]. Although the 5-year survival rate of PCa is high when detected early, advanced stages are very aggressive and difficult to treat, including resistance to castration. While prostate-specific antigen (PSA) levels are commonly used for PCa detection, they may also be elevated due to other conditions, such as prostatitis [42]. Positive PSA test results may lead to an invasive biopsy, which makes it crucial to develop a more conclusive noninvasive test to reduce unnecessary procedures. HP [1-^13^C] pyruvate MRSI offers an elegant solution to detect PCa, identify metastatic tumors, and monitor treatment response. Preclinical research has identified the lactate-to-pyruvate ratio as an effective tool in interrogating PCa metabolic pathways. HP [1-^13^C] lactate generation from pyruvate may distinguish between normal prostate and cancers categorized as low- or high-grade on the basis of histology in the transgenic adenocarcinoma of mouse (TRAMP) prostate cancer model [43]. In different investigations using patient-derived prostate tissue slices built into perfused bioreactor systems, malignant tissue slices converted pyruvate to lactate at a considerably greater rate than benign prostate tissue [44]. Higher pyruvate-to-lactate conversion was linked to increased expression of the transporters MCT1 and MCT4, as well as the enzyme LDHA. Pyruvate-to-lactate conversion rates can be used for in vivo phenotyping of androgen receptor (AR) expression, which affects PCa growth and metabolism. Increased lactate production was observed in AR-dependent castration-resistant PCa models compared to the more aggressive variant with loss of AR expression [45]. In a separate study, PCa cell lines with different metabolic phenotypes and varying levels of aggressiveness were tested using HP [1-^13^C] pyruvate MRSI. This approach has shed light on the nuanced relationship among MCT1 expression level, LDH isoform ratio, and the regulation of pyruvate-to-lactate conversion in cancer cells [46]. In addition, HP [1-^13^C] pyruvate MRSI can allow for the critical assessment of the PCa model’s responses to various therapies. Another study used the pyruvate-to-lactate conversion ratio to evaluate the effectiveness of glycolysis targeting in various PCa cell line xenografts. However, it is worth noting that steady-state assessments may not be sufficient for accurately assessing LDH activity [47]. In patients, HP [1-^13^C] pyruvate MRSI has been shown to distinguish between healthy tissue and PCa via the ^13^C-labeled lactate-to-pyruvate ratios (Figure 3) [11]. Recently, pilot studies yielded promising results in the use of HP [1-^13^C] pyruvate MRSI to detect bone and liver metastases through the use of the lactate conversion rate [48]. HP [1-^13^C] pyruvate MRSI has also recently been investigated to gauge tumor response to androgen deprivation therapy, in which metabolic changes in the lactate-to-pyruvate ratios appear before the morphologic response to therapy [10,45].

## 6. Breast Cancer

Breast cancer is one of the leading causes of cancer-related deaths in women, primarily due to its significant heterogeneity. This heterogeneity is evident within and across tumors and is often linked to variations in hormone receptor expression and HER2 amplification/overexpression. In a study using a perfused bioreactor system, it was discovered that the conversion of HP [1-^13^C] pyruvate to lactate in breast cancer cells was mechanistically linked to the expression of MCT1 (monocarboxylate transporter-1) [49]. Subsequently, another study conducted in a doxycycline-inducible MYC-driven murine breast cancer model found a correlation between the generation of lactate from HP [1-^13^C] pyruvate and the development or regression of MYC-driven tumors [50]. The use of HP [1- ^13^C] pyruvate MRSI to evaluate early response to neoadjuvant chemotherapy was demonstrated in human breast cancer subjects [14]. A 34% drop in the ^13^C-labeled lactate-to-pyruvate ratio after one round of neoadjuvant chemotherapy resulted in the proper designation of the patient as a responder to treatment, which was later confirmed by pathological evidence [14]. In another study with human breast cancer patients, lactate labeling linked with MCT1 expression and hypoxia, as well as imaging of HP [1-^13^C] pyruvate metabolism, in breast cancer indicated considerable intra- and intertumoral metabolic variability [15]. In this paper, a lack of correlation of malignancy with LDH activities was reported. This is corroborated by studies with breast cancer xenografts in vivo [51] and cell lines in vitro [52]. Such a lack of correlation between pyruvate-to-lactate conversion and malignancy was also reported for hepatocellular carcinoma [53]. In fact, emerging quantitative studies have now demonstrated that hyperpolarized [1-^13^C] pyruvate MRSI measures primarily MCT1-mediated [1-^13^C] pyruvate transmembrane influx in vivo, not glycolytic flux or LDHA activity, driving a reinterpretation of this technology during clinical translation [54,55]. Therefore, the general concept of hyperpolarized pyruvate-to-lactate conversion being correlated with LDH expression and (total) activity as indicators of malignancy is not stringent, and this is an important message for the application and interpretation of this methodology.

## 7. Liver Cancer

Liver cancer is known as one of the leading causes of cancer-related deaths globally. One way to detect cancer in liver is using HP [1-^13^C] pyruvate MRSI. The development of hepatocellular carcinoma (HCC) typically occurs in stages, starting with the formation of regenerative nodules, before progressing to dysplastic nodules, then to early HCC, and ultimately to malignant HCC [56,57]. In one of the HP [1-^13^C] pyruvate MRSI studies, precancerous regions in the *MYC* gene-driven mouse model showed upregulated flux of pyruvate to alanine compared to normal and cancerous cohort mouse models [58]. In another study of hepatocellular carcinoma-bearing rats, tumors could be detected with integrated images of [1-^13^C] lactate MRSI and ^18^F-FDG PET [47]. It was found that a hyperpolarized signal is not sufficient for spatially and temporally resolved ^13^C MRSI, emphasizing the need to develop new MR acquisition methods [53]. An interesting study on buffalo rats bearing orthotopic HCC showed higher pyruvate-to-alanine conversion than pyruvate-to-lactate conversion, suggesting a potential biomarker to diagnose HCC [59]. The presence of significantly upregulated alanine transaminase was observed in HCC tumor tissues, correlating with higher alanine conversion. However, in a nude mouse model where tumors were located in the flank, higher lactate conversion from pyruvate was observed instead of alanine. This discrepancy may have been due to the fact that the subcutaneous mouse model used in this study did not accurately reflect the microenvironment of the liver, and the isolated cancer cells injected into nude mice may have undergone cell culture-related selection processes [60]. The hypometabolic conditions due to trans-arterial embolization (TAE) in hepatocellular tumors in rats were identified with HP [1-^13^C] pyruvate MRSI and showed a reduced alanine-to-lactate ratio compared to pre-TAE rats and control cohorts [61]. These results are in agreement with histology observations.

A real-time assessment of metabolism using HP [1-^13^C] pyruvate MRSI on prostate cancer metastases to the liver and bone in human patients revealed that HP [1-^13^C] pyruvate MRSI can be used to diagnose not only localized cancer, but also metastatic cancers [62]. The mean pyruvate-to-lactate *k_PL_* values were 0.020 ± 0.006 (s^−1^) in bone and 0.026 ± 0.000 (s^−1^) in liver. Furthermore, the HP [1-^13^C] pyruvate MRSI study after 2 months of treatment with taxane + platinum chemotherapy showed the efficacy of treatment with a reduced *k_PL_* value of 0.015 (s^−1^). This clinical study supports the feasibility of HP [1-^13^C] pyruvate MRSI in future clinical studies of metastatic cancer [62,63]. 

## 8. Gastric Cancer

Gastric cancer is on the rise globally and is the third leading cause of cancer-related deaths worldwide. Extensive research has been conducted on the metabolomics, genomics, transcriptomics, and proteomics of gastric cancer with the aim of identifying biomarkers that aid in early detection [64]. In another study, gastric cancer nude mouse models bearing NCI-N87 tumors were randomly segregated into two groups and were either treated with afatinib (pan-tyrosine kinase inhibitor) or treated with vehicle. The mouse model did not show significant differences on day 0 and day 4 in PET uptake in afatinib-treated groups, as well as in vehicle-treated groups. However, HP [1-^13^C] pyruvate MRSI showed significant differences in the lactate-to-pyruvate ratio on day 0 and day 4 of the afatinib-treated group, whereas the lactate-to-pyruvate ratio remained the same in the vehicle-treated group on day 0 and day 4 after treatment. These results indicated that HP [1-^13^C] pyruvate MRSI can capture the early efficacy of drugs before morphological changes occur [65]. One of the studies developed gastric cancer tumors spontaneously by overexpressing peroxisome proliferator-activated receptor delta (PPARD) in villin-positive gastric progenitor cells. These mice were studied to identify the metabolic pathways fueling cancer tumor growth. The ex vivo tumor tissues were subjected to NMR spectroscopy and LC–MS-based metabolic studies. Ten-week-old and 35-week-old PPARD mice were used to study the role of glycolysis in tumorigenesis using HP [1-^13^C] pyruvate MRSI. The HP [1-^13^C] pyruvate MRSI study showed no significant difference in the pyruvate-to-lactate flux in 10-week-old and 35-week-old PPARD mice. The NMR study found significantly altered concentrations of inosine monophosphate (*p* = 0.0054), adenosine monophosphate (*p* = 0.009), UDP-glucose (*p* = 0.0006), and oxypurinol (*p* = 0.039) as PPARD mice aged from 10 weeks to 35 weeks and 55 weeks. LC–MS studies showed decreased concentrations of palmitic acid (*p* = 0.0029), oleic acid (*p* = 0.0007), steric acid (*p* = 0.0028), and linoleic acid (*p* = 0.0015) in 55 week old PPARD mice compared to 10 week old PPARD mice [66]. These results suggested that the gastric cancer tumor proliferation and energy fueling in PPARD-driven gastric cancer is dependent on fatty acids rather than glycolysis [66].

## 9. Melanoma

Melanoma has been investigated for early diagnosis and therapeutic intervention using HP [1-^13^C] pyruvate MRSI. BRAF is one of the genes upregulated in melanoma cells, and inhibition of BRAF signaling has been a top target for therapy. Metabolic consequences as a result of the inhibition of BRAF by vemurafenib have been investigated using HP [1-^13^C] pyruvate MRSI in melanoma cell lines [67,68,69]. One of the studies showed a reduction in pyruvate-to-lactate conversion in the BRAF mutant melanoma cell line compared to the vehicle control, whereas an increased pyruvate-to-lactate conversion was recorded in the BRAF wildtype melanoma cell line compared to vehicle control [67]. In another study, the combined treatment of BRAF/MEK inhibitors vemurafenib and trametinib in melanoma xenografts showed no significant changes in lactate-to-pyruvate ratios before treatment and after 24 h of combination treatment. On the other hand, in response to vemurafenib and trametinib, ex vivo ^13^C steady-state metabolic studies showed significantly reduced ^13^C-lactate production from U-^13^C-glucose. However, 5-^13^C-glutamine metabolism was not altered in response to vemurafenib and trametinib [68]. The patient-derived xenografts were treated with the BRAF inhibitor vemurafenib and vehicle. The effect of BRAF inhibitor was tested using HP [1-^13^C] pyruvate MRSI, and the results demonstrated an increased pyruvate-to-lactate conversion in vemurafenib-treated animals compared to vehicle-treated mice. The increased pyruvate-to-lactate conversion was attributed to tumor microenvironment influence [65,69].

Immune check point blockade therapy works by blocking the proteins which negatively control the T-cell action. The import proteins of target in ICB are cytotoxic T-lymphocyte antigen-4 (CTLA-4), programmed death-1 (PD-1), and programmed death ligand 1 (PDL-1) [65,66]. The HP [1-^13^C] pyruvate MRSI was used to assess the immune checkpoint blockade (ICB) therapy. Multimodal imaging of HP [1-^13^C] pyruvate, HP [1,4–^13^C2] fumarate MRSI, and dynamic contrast-enhanced (DCE) MRI was used to assess the ICB in MC38 colon adenocarcinoma and B16-F10 melanoma. The study found that MC38 colon adenocarcinomas were more sensitive to ICB therapy, whereas B16-F10 melanomas were less sensitive to therapy according to the [1,4–^13^C2] fumarate study. More necrosis in MC38 colon adenocarcinoma was found from a higher fumarate-to-malate conversion compared to vehicle control. The combined imaging modalities HP [1-^13^C] pyruvate, HP [1,4–^13^C2] fumarate MRSI, and dynamic contrast-enhanced (DCE) MRI can provide noninvasive imaging biomarkers to find the early responses to ICB [70]. However, one limitation of the study was the subcutaneous mouse model, which may not represent a similar biology to orthotopic mouse models. In another study, the ICB resistance in a BL6/B16 melanoma mouse model was investigated using HP [1-^13^C] pyruvate MRSI, as well as other biochemical and biophysical experiments. The ICB-resistant model was developed through an in vivo series of cell line passages and ICB treatment. The completely resistant model was called F4 BL6/B16, and the responding one was referred to as TMT. The HP [1-^13^C] pyruvate MRSI found that the resistant F4 BL6/B16 mice showed significantly higher pyruvate-to-lactate conversion compared to the ICB-responding TMT BL6/B16 mice. The study concluded that ICB resistance tumors acquired hypermetabolic phenotypes with coordinated upregulation of glycolysis, oxidoreductase, and mitochondrial oxidative phosphorylation. This study was promising as it showed that the lactate-to-pyruvate ratio can serve as a noninvasive biomarker to identify immunotherapy resistance in melanoma [71].

## 10. Renal Cell Carcinoma

Hyperpolarized [1-^13^C] pyruvate techniques have proven effective in imaging a range of cancer types, including renal cell carcinoma (RCC) and leukemia, in addition to those previously discussed. Specifically, in preclinical models of renal cell carcinoma, HP [1-^13^C] pyruvate has demonstrated the ability to predict aggressiveness. The identification of RCCs has grown as access to clinical MRI has increased widely in recent years. MRI, on the other hand, cannot distinguish between low-grade, indolent RCCs that may be handled with active surveillance and high-grade aggressive RCCs that require surgery. Other study demonstrated that the HP lactate generation and efflux distinguished between normal renal epithelial cells, localized RCC, and metastatic RCC, allowing for tumor detection [13]. Other study demonstrated that HP [1-^13^C] pyruvate MRSI can be used to investigate tumor lactate production and compartmentalization noninvasively in murine orthotopic models of human RCCs [72]. This technique also provides insight into tumor LDHA and MCT4 expression, which have been linked to tumor aggressiveness [72].

## 11. Leukemia

For nonsolid tumors such as leukemia, it is difficult to use HP [1-^13^C] pyruvate for metabolic imaging, but there are a couple of reports indicating that it is indeed possible. Instead of focusing on the tumor or bloodstream they focus on the bone marrow, where metabolic reprogramming is suggested to occur along with a transition to a more hypoxic environment. Other study demonstrated that, as leukemia progresses in the acute myeloid leukemia (AML) mouse model, the hypoxic conditions increase, and an observable shift in glycolysis occurs. This was observed by an increase in the lactate-to-pyruvate ratio following the injection of HP [1-^13^C] pyruvate [73]. The bone marrow was imaged in an AML-model before and after treatment with a glutaminase inhibitor, CB-839. They indicated that the lactate-to-pyruvate ratio decreased after therapeutic intervention. To validate the in vivo results, they employed HP [1-^13^C] pyruvate in vitro with AML cells, and they observed similar results [74]. This indicates that HP [1-^13^C] pyruvate can be employed to observe metabolic changes after treatment in nonsolid tumors. Table 1 gives a brief overview of research articles in the field of the application of pyruvate [1-^13^C] HP-MRSI in diagnosis and therapeutic intervention in different cancer systems.

## 12. Future Directions

Hyperpolarized metabolic imaging has made significant progress over the past decade, moving from preclinical applications to clinical settings in cancer and cardiovascular diseases. However, there are still many areas of research that can benefit from the application of hyperpolarization [6,75,76]. One of the emerging frontiers is to combining multiple metabolic imaging techniques such as acidoCEST MRI, PET, electron paramagnetic resonance imaging (EPRI), and mass spectrometry imaging (MSI) with hyperpolarized metabolic imaging of different compounds (pyruvate and beyond) to get a more detailed and mechanistic understanding on different cancer systems combined with deep-tissue imaging. Another area of great contemporary interest is the application of hyperpolarization to interrogate and classify the metabolism of different microbiome systems in cancer [77,78]. As more is learned about the relationship between the microbiome and cancer, hyperpolarized metabolic imaging may become an important tool for understanding how the microbiome influences cancer development and progression.

Lastly, there is a growing interest in leveraging artificial intelligence (AI) and HP MRSI applications together to develop real-time, actionable biomarkers for early detection, assessing aggressiveness, and interrogating the early efficacy of therapies in many cancer systems [30]. For example, multimodal AI can learn features from HP MRSI, as well as standard-of-care anatomical MRI, PET, and CT imaging modalities, to yield “hybrid biomarkers” and reduce the time required to detect evolution in cancers [79,80,81]. This approach can help clinicians make more informed decisions about treatment options and may ultimately lead to better outcomes for cancer patients. Overall, the future of hyperpolarized metabolic imaging looks promising, with many exciting directions for research and clinical applications.

However, there are still some challenges that need to be addressed before the technology can be fully utilized in clinical practice. One of the primary challenges is the quantification of the injected pyruvate in tumor cells and its subsequent conversion to lactate. Additionally, the spatial resolution of pyruvate and lactate in imaging poses a challenge, and contribution of the tumor microenvironment in the pyruvate-to-lactate flux needs to be better understood [82]. Another important challenge is the consistency in obtaining hyperpolarized signals across experiments, which can affect the accuracy and reproducibility of the results. Addressing these challenges through continued research and development and multicenter clinical trials is crucial for the full realization of the potential of this technology in clinical practice.

## 13. Conclusions

The review provided an insightful overview of the applications of HP [1-^13^C] pyruvate MRSI in different types of cancers, including preclinical and clinical studies. As most of the cancers have altered glucose metabolism, HP [1-^13^C] pyruvate MRSI will be the prioritized diagnosis choice in the future because of its advantages of being radiation-free and providing information on both uptake and downstream metabolic fate. The technology has shown great potential in aiding early diagnosis and therapeutic intervention for cancer patients in the near future. This review focused only on metabolite HP [1-^13^C] pyruvate, but several other molecules (glutamine, acetate, carnitine etc.) involved in different biochemical pathways also have the potential to diagnose and interrogate the therapeutic intervention of cancer. Overall, HP [1-^13^C] pyruvate MRSI has demonstrated significant potential in advancing cancer diagnosis and treatment. With further research and development, this technology could become an essential tool in the management of cancer patients worldwide.

## Figures and Tables

**Figure 1 metabolites-13-00606-f001:**
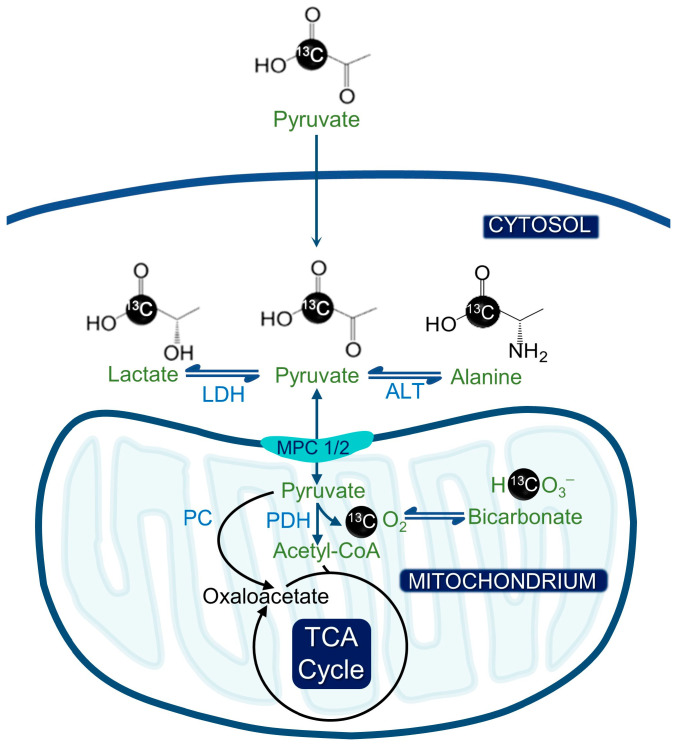
Hyperpolarized pyruvate metabolism and its significance for metabolic imaging in cancer. The hyperpolarized [1-^13^C] pyruvate can be converted to [1-^13^C] alanine or [1-^13^C] lactate by alanine transferase (ALT) or lactate dehydrogenase (LDH). Alternatively, [1-^13^C] pyruvate can be converted to acetyl CoA by pyruvate dehydrogenase (PDH), whereby the ^13^C label is lost as CO_2_ and ultimately bicarbonate.

**Figure 2 metabolites-13-00606-f002:**
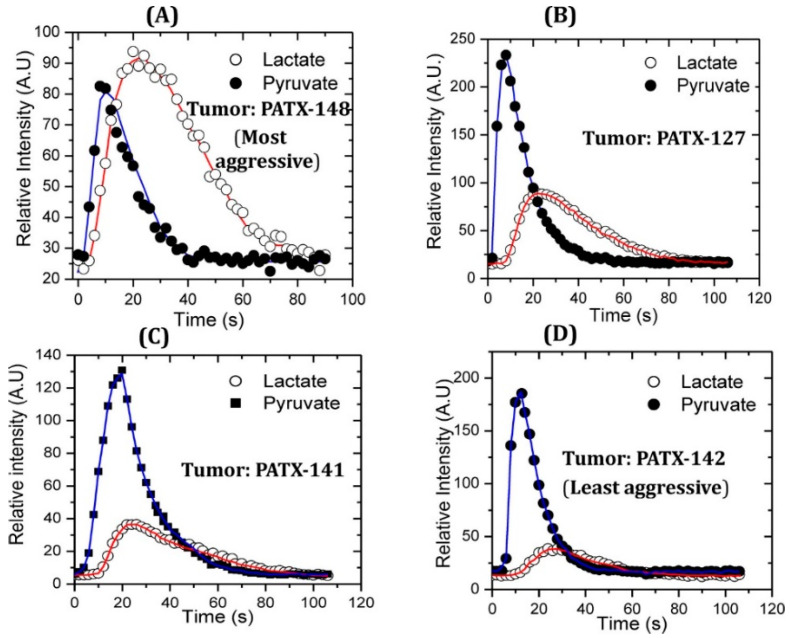
Higher pyruvate-to-lactate conversion is associated with more aggressive pancreatic cancer in a comparative study with patient-derived murine xenograft models of pancreatic cancer. Patient-derived pancreatic cancer xenograft PATX-148 (**A**) is the most aggressive cancer which shows higher lactate flux compared to moderately aggressive PDX pancreatic cancer models PATX-127 (**B**) and PATX 141 (**C**). Slow growing and the least aggressive PDX model PATX-142 (**D**) compared in this study showed the lowest pyruvate-to-lactate conversion. Red line connects the signals of lactate obtained over time and blue line connects the signals of pyruvate obtained over time. Reproduced with a permission from ACS publications Ref. [31].

**Figure 3 metabolites-13-00606-f003:**
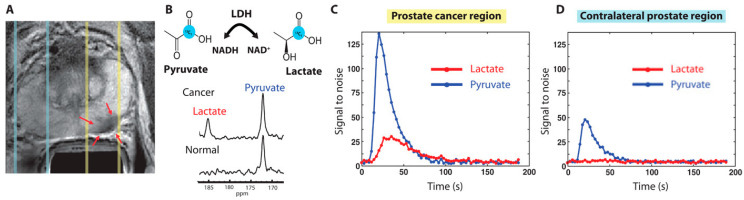
(**A**) Proton T2 weighted MRI from a patient with confirmed prostate cancer from PSA levels of 12.2 ng/mL and biopsy. Red arrows under yellow dashed lines are biopsy proven areas and the zone under blue dashed lines are normal prostate areas. (**B**) HP-MR spectrum was obtained with injection of HP [1-^13^C] pyruvate. Cancer regions produced the lactate of SNR 25 from HP [1-^13^C] pyruvate, proving the presence of cancer. (**C**,**D**) Localized 1D hyperpolarized pyruvate and lactate plotted as a time interval. The slice area covering the prostate tumor has more pyruvate uptake and as well lactate conversion (**C**) compared to the contralateral prostate region (**D**). The image has been reproduced with permission from the American Association for the Advancement of Science publishers Ref. [11].

**Table 1 metabolites-13-00606-t001:** The applications of [1-^13^C] pyruvate HP-MRSI in diagnosis and therapeutic intervention in different cancer systems is summarized in the Table 1 with references ([Ref]).

Cancer Type	Model	Diagnosis/Therapeutic Intervention/Mechanism	Lactate/Pyruvate	References
Brain	[12]. Human	[12]. Evaluation of in vivo brain metabolism imaging in patients.	[12]. Decreased	[12,20,21,22,23,24,25,26,27,28]
	[20]. Human xenograft on rat	[20]. Distinction between malignant glioma tissue and normal tissue. Observation of differences between U-251 MG and U-87 MG models.	[20]. Increased	
	[21]. Cell line	[21]. Distinction between lower grade gliomas with IDH1 mutation and glioblastoma.	[21]. Decreased	
	[22]. Rat	[22]. Detection of pyruvate dehydrogenase flux modulated by dichloroacetate.	[22]. Increased	
	[23]. Rat	[23]. Measure of treatment response to whole brain irradiation.	[23]. Decreased	
	[24]. Rat	[24]. Measure of response to Everolimus treatment.	[24]. Decreased	
	[25]. Murine model	[25]. Determination of pseudoprogression after therapy.	[25]. Increased	
	[26]. Human	[26]. Comparison of metabolism between untreated and recurrent tumors.	[26]. Increased	
	[27]. Cell line; Rat	[27]. Measure of disease progression by imaging tumors with c-Myc expression, which correlated with tumor grade.	[27]. Increased (cell line); Decreased (rat)	
	[28]. Rat	[28]. Quantification of tumor metabolic profile corroborated with histopathology.	[28]. Increased	
Pancreas	[16]. Human	[16]. Characterization of pancreatic cancer heterogeneity and hypoxia.	[16]. N/A	[16,31,32,33,34,35,36,37]
	[31]. Human xenograft on mouse	[31]. Relation between abnormal glycolytic metabolism and tumor progression.	[31]. Increased	
	[32]. Mouse	[32]. Detection of pancreatic preneoplasia prior to metastasis.	[32]. N/A	
	[33]. Mouse	[33]. Detection of pancreatic intraepithelial neoplasias which aids earlier detection of pancreatic cancer.	[33]. Increased	
	[34]. Human xenograft on mouse	[34]. Assessment of response to hypoxia-activated prodrugs.	[34]. N/A	
	[35]. Mouse	[35]. Assessment of response to LDH-A inhibitors.	[35]. Decreased	
	[36]. Cell line	[36]. Investigation of Quinone Oxidoreductase 1 mediated redox cycle.	[36]. Decreased	
	[37]. Human	[37]. Prediction of radiation therapy response.	[37.] N/A	
Ovary	[39]. Mouse	[39]. Assessment of response to multityrosine kinase inhibitor treatment.	[39]. Increased	[39]
Prostate	[10]. Human	[10]. Investigation of tumor response to androgen deprivation therapy.	[10]. Decreased	[10,11,43,44,45,46,47,48]
	[11]. Human	[11]. First-in-man study which distinguishes normal and cancerous tissue.	[11]. Increased	
	[43]. Mouse	[43]. Correlation of lactate/pyruvate ratio with histologic grades.	[43]. Increased	
	[44]. Tissue slice culture	[44]. Interrogation of glucose reprogramming.	[44]. Decreased	
	[45]. Human xenograft on mouse	[45]. Phenotype identification of androgen receptors, MCT1, MCT4, and LDHA expression.	[45]. Increased	
	[46]. Cell line	[46]. Investigation of relationship between MCT1 expression, LDH isoform ratio, and regulation of glycolysis.	[46]. Increased	
	[47]. Human xenograft on mouse	[47]. Evaluation of glycolysis targeting efficacy.	[47]. Decreased	
	[48]. Human	[48]. Detection of bone and liver metastases.	[48]. Increased	
Breast	[14]. Human	[14]. Assessment of early response to neoadjuvant chemotherapy.	[14]. Decreased	[14,15,49,50,51,52]
	[15]. Human	[15]. Demonstration of tumor metabolic heterogeneity, MCT1 expression, and hypoxia.	[15]. Increased	
	[49]. Cell line	[49]. Investigation of MCT1 and malignant transformations.	[49]. Increased	
	[50]. Murine model	[50]. Identification of correlation between glycolytic activity and tumor regression.	[50]. Decreased	
	[51]. Human xenograft on mouse; Cell line	[51]. Correlation of LDH activity and metastatic potential of tumors.	[51]. Increased (human xenograft on mouse); Increased (cell line)	
	[52]. Cell line	[52]. Investigation of glucose and glutamine availability and corresponding effects on tumor metabolism.	[52]. Varied based on condition	
Liver	[53]. Rat	[53]. Detection of glycolytic activity in tumor tissue.	[53]. Increased	[53,58,59,60,61,62]
	[58]. Mouse	[58]. Investigation of glycolytic processes involved in tumor formation and regression.	[58]. Decreased	
	[59]. Rat	[59]. Identification of new biomarkers for diagnosis.	[59]. N/A	
	[60]. Mouse	[60]. Assessment of different tumor phenotype and corresponding metabolic profile.	[60]. Increased	
	[61]. Rat	[61]. Identification of hypometabolic conditions.	[61]. N/A	
	[62]. Human	[62]. Quantification of early treatment response in metastases.	[62]. Decreased	
Gastric	[65]. Cell line; Mouse	[65]. Assessment of early treatment response to tyrosine kinase inhibitor therapy.	[65]. Decreased or no change (cell line); Decreased (mice)	[65,66]
	[66]. Mouse	[66]. Investigation of metabolic pathways involved in tumorigenesis.	[66]. No change	
Melanoma	[67]. Cell line	[67]. Investigation of BRAF inhibition responses and metabolic effects.	[67]. Decreased	[67,68,69,70,71]
	[68]. Human xenograft on mouse	[68]. Investigation of BRAF/MEK inhibition responses.	[68]. No change	
	[69]. Human xenograft on mouse	[69]. Assessment of early treatment response to BRAF inhibition.	[69]. Increased	
	[70]. Mouse	[70]. Assessment of response to immune checkpoint blockade therapy in a noninvasive way.	[70]. Decreased or no change	
	[71]. Mouse	[71]. Investigation of the underlying metabolic pathways for checkpoint blockade resistance.	[71]. Increased	
Renal Cell	[13]. Human	[13]. Prediction of renal cell carcinoma aggressiveness.	[13]. N/A	[13,72]
	[72] Mouse	[72]. Investigation of tumor aggressiveness based on lactate production, LDHA expression, and MCT4 expression.	[72]. Increased	
Leukemia	[73]. Murine model	[73]. Investigation of leukemia metabolic pathways and hypoxia.	[73]. Increased	[73,74]
	[74]. Cell line	[74]. Assessment of acute myeloid leukemia treatment response to glutaminase inhibitor.	[74]. Decreased	

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
