# Peer review of "Enhancing Cancer Diagnosis with Real-Time Feedback: Tumor Metabolism through Hyperpolarized 1-13C Pyruvate MRSI"

_metabolites, 2023, doi:10.3390/metabo13050606_

Round 1

Reviewer 1 Report

Manuscript Review

Journal: Metabolites

Manuscript ID: metabolites-2313773

Manuscript Type: Review

Manuscript Title: Revolutionizing Cancer Diagnosis: A Closer Look at Tumor Metabolism Through Hyperpolarized 1-13C Pyruvate MRSI.

General Comments:

This manuscript provides a near exhaustive review of the cancer-focused literature related to hyperpolarized (HP) carbon-13 (13C) magnetic resonance spectroscopic imaging (MRSI), principally that employing the pyruvate substrate labeled at carbon #1 [1-13C]pyruvate. The manuscript will be of value to those in the HP MR field and those interested in learning more about the emerging arena of HP MR, especially cancer applications.  The manuscript’s writing is generally clear and suitably concise, and its citations are appropriately extensive.  This reviewer has only a few minor suggestions toward possible improvements.

Specific Comments:

1)    The manuscript’s title is a bit hyperbolic.  “Revolutionizing”, really?  Perhaps “Innovating” or “Enhancing” or “Augmenting” or “Improving” would lower the marketing temperature sufficiently.

2)    Page 2, Line 78.  Remove second occurrence of “clinical studies”.

3)    Page 6, Lines 222 & 223.  Suggest rewriting as: “Prostate cancer (PCa) is the second most common cancer-caused fatality among men [40, 41].”

4)    Page 7, Line 301.  Suggest replacing “claiming the potential biomarker” with “suggesting a potential biomarker”.

5)    Page 8, Line 315.  Suggest replacing “but also metastases cancers” with “but also metastatic cancers”.

6)    Page 8, Line 322.  Suggest replacing “Gastric cancer events are on the rise” with Gastric cancer is on the rise”.

7)    Page 8, Lines 338 and 341:  Suggest “Ten-week and 35-week old PPARD mice” and “10-week and 35-week old PPARD mice”.

8)    Page 9, Line 370.  Suggest deleting the word “The” at the start of the sentence.

Author Response

We are very thankful for reviewer for his suggestions and corrections. The changes can be tracked in revised manuscript.

1)    The manuscript’s title is a bit hyperbolic.  “Revolutionizing”, really?  Perhaps “Innovating” or “Enhancing” or “Augmenting” or “Improving” would lower the marketing temperature sufficiently.

Ans: The title has been changed to –“ Enhancing Cancer Diagnosis with Real-Time Feedback: Tumor Metabolism Through Hyperpolarized 1-13C Pyruvate MRSI.”

2)    Page 2, Line 78.  Remove second occurrence of “clinical studies”.

Ans: Removed

3)    Page 6, Lines 222 & 223.  Suggest rewriting as: “Prostate cancer (PCa) is the second most common cancer-caused fatality among men [40, 41].”

Ans: Corrected as suggested

4)    Page 7, Line 301.  Suggest replacing “claiming the potential biomarker” with “suggesting a potential biomarker”.

Ans: Corrected as suggested

5)    Page 8, Line 315.  Suggest replacing “but also metastases cancers” with “but also metastatic cancers”.

Ans: Corrected as suggested

6)    Page 8, Line 322.  Suggest replacing “Gastric cancer events are on the rise” with Gastric cancer is on the rise”.

Ans: Changed as suggested

7)    Page 8, Lines 338 and 341:  Suggest “Ten-week and 35-week old PPARD mice” and “10-week and 35-week old PPARD mice”.

Ans: Corrected as suggested

8)    Page 9, Line 370.  Suggest deleting the word “The” at the start of the sentence.

Ans: Deleted as suggested

Reviewer 2 Report

Nice review article on13C-pyruvate MRS imaging application for metabolic diagnostics in cancer patients for better therapeutics.

A few more points authors should include and describe better,

  1. How C13-labelled pyruvate is converted to other metabolites and how labelled C13 gets transferred to other metabolites.
  2. Please redraft Figure 1 with more information about C13 pyruvate labeling.
  3. What are the advantages and disadvantages of this MRSI technique compared to the LC-MS/MS and GC approaches?

Author Response

We thank reviewer for his comments and suggestions to improve the manuscript.

1) How C13-labelled pyruvate is converted to other metabolites and how labelled C13 gets transferred to other metabolites.

Ans: The metabolites converted from 1-13C labeled pyruvate to lactate, alanine, and carbon dioxide in the presence of enzymes-pyruvate dehydrogenase, alanine dehydrogenase, and pyruvate decarboxylase respectively. The pyruvate fluxes can be tracked in real time using pyruvate [1-13C] HP-MRSI in in vivo and by 13C NMR spectroscopy and mass spectrometry in vitro. However, to track the citric acid metabolites from pyruvate, 2nd carbon of the pyruvate has to be labeled.  

2) Please redraft Figure 1 with more information about C13 pyruvate labeling.

Ans: Incorporated suggestions in Figure 1

3) What are the advantages and disadvantages of this MRSI technique compared to the LC-MS/MS and GC approaches?

Ans: Pyruvate [1-13C] HP-MRSI is non-invasive metabolic imaging technique where the flux of the metabolites can be tracked by 13C labeling in real time. This technique provides complementary information over LC-MS/MS and GC which are destructive analytical techniques. In LC-MS/MS the tissues will be collected by biopsy and subjected to mass spectroscopic study to get information on metabolomics. LC-MS/MS and GC gives static pictures of metabolomics and metabolic pool sizes, whereas pyruvate [1-13C] HP-MRSI provides a dynamic picture of metabolism of tumor tissues as the biochemical event is unfolding in real-time. Mass spectroscopy has an advantage of detecting higher number of metabolites compared to NMR and MRSI and requires a smaller amount of tissues compared to NMR spectroscopy-based metabolomics.

Reviewer 3 Report

The manuscript is well structured, comprehensive, and in summary an interesting read.
Future directions could be extended, though AI has not been leveraged on hyperpolarized C13mrsi yet, the topic of AI could be mentioned in more detail, s.a. https://doi.org/10.1007/s00247-021-05177-7, https://doi.org/10.1007/s00345-022-03930-7, https://doi.org/10.1016/j.ejmp.2021.03.003, including possible prospective BraTS (Brain Tumor Segmentation) challenges e.g. or similar iniatives.
Typos p.10 l.419, p.10 l.436.

Author Response

1) Future directions could be extended, though AI has not been leveraged on hyperpolarized C13mrsi yet, the topic of AI could be mentioned in more detail, s.a. https://doi.org/10.1007/s00247-021-05177-7, https://doi.org/10.1007/s00345-022-03930-7, https://doi.org/10.1016/j.ejmp.2021.03.003, including possible prospective BraTS (Brain Tumor Segmentation) challenges e.g. or similar initiatives.

Ans: We thank the reviewer for this addition.  The references has been added.

2) Typos p.10 l.419, p.10 l.436.

Ans: Corrected the typos.

Reviewer 4 Report

In the current review, the authors focus on the use of HP [1- 13C] pyruvate MRSI as a more representative biomarker of early metabolic changes in cancer that could be used for early prediction of response to targeted therapies. The study is interesting. The paper is very well written, well organized and fits in the scope of the journal. The introduction justifies the aim of the study; the data in the context of existing literature highlight the significance of the method and the possible impact in clinical practice, In the conclusion, the authors refer to challenges that need to be addressed before this method can be fully utilized and they also set the need for further studies.

Minor comment: A table summarizing the type of cancer and the efficiency of this new technology would be very useful.

Author Response

We thank reviewer for his/her suggestions.

1) Minor comment: A table summarizing the type of cancer and the efficiency of this new technology would be very useful

Ans: Incorporated the table summarizing the HP-MRSI application in different cancer systems

Reviewer 5 Report

The authors discuss the potential of hyperpolarized (HP) 13C magnetic resonance spectroscopic imaging (MRSI) as a noninvasive technique for identifying altered metabolism in various cancer types. This is a comprehensive and technically competent review of previous experimental work. The subject is of interest in the field of metabolic imaging technology. In my opinion, the paper is generally well-written and structured. I recommend the article for publication.

Author Response

1) The authors discuss the potential of hyperpolarized (HP) 13C magnetic resonance spectroscopic imaging (MRSI) as a noninvasive technique for identifying altered metabolism in various cancer types. This is a comprehensive and technically competent review of previous experimental work. The subject is of interest in the field of metabolic imaging technology. In my opinion, the paper is generally well-written and structured. I recommend the article for publication.

Ans: We thank reviewer for his/her encouraging and positive feedback.

Reviewer 6 Report

Comments to the authors:

Summary

Advancements in the technology of administering and analyzing the conversions of hyperpolarized [1-13C]pyruvate for metabolic dynamics and imaging have led to numerous preclinical and exemplary clinical studies. In this timely review article, the authors have collected a number of reports illustrating the applications in diagnosis and treatment response in different types of cancers.

The manuscript is well structured in focusing on different cancer types in individual sections, and in highlighting both in-vitro and in-vivo studies as well as first patient applications. The numerous references on this methodology illustrate the growing interest and recent developments.

Comments on the manuscript

General

In spite of the wealth of information presented in the manuscript, there are some weaknesses in the text which blur the understanding of the context and make it difficult to extract the gain of knowledge. The manuscript needs editing and more diligence in conveying the essential points, which are often obscured by unnecessary details: i.e. many parts of this manuscript are not "succinct".  Some specific recommendations:

a)      In the Title: "Revolutionizing" ... is an overstatement (not suitable in a scientific paper). Surely, the authors have a more appropriate word. (suggestion: New Potentials)

b)      When describing the contents of referenced publications, do avoid naming first authors, as this does not add any relevant information and disturbs the sentence flow and context.

c)      Also, check for redundancies within a sentence (examples: lines 176/179; 331/2; 372/378)

d)       Some paragraphs read more like a collection of reports and lack a narrative. Check for a main thought and a concluding message in each paragraph.

e)      Avoid abbreviations where possible, as some are very specific to a field in cancer research and not generally familiar. Instead, use a short word /or explanatory term to highlight the relevance of a tumor type (model). However, the abbreviation LDH for lactate dehydrogenase is of common use in cancer-related terminology.

f)       Figure 1 needs revision/amendments: The legend describes hyperpolarized [1-13C]pyruvate. This should also be depicted in the figure. The potential conversion products should also be 13C-labeled and would explain the appearance of 13CO2 from entry of 13C-pyruvate-derived acetyl-CoA in to the oxidative TCA cycle.  Also, the path of carboxylation from [1-13C] pyruvate would result in 13C-oxaloacetate and eventually 13C-malate. PC needs to be spelled out.

g)      Figures 2 and 3 – They should be omitted, as they are already published as presented, are too detailed here, and need more explanations for interpretation, which would be out of the scope of this overview. More important here would be to highlight the relevant results and conclusions. Readers interested in the details could consult the referenced publication.

2. Brain cancers

a)      Paragraphs 1 and 2 need better phrasing of the lines of thought, i.e. intentions and the message. In line 91: what are the "unique characteristics" relevant to the context?

b)      Line 111: define what is meant by "pseudoprogression"

3. Pancreatic Cancers

a)      This section needs a better structure, i.e. more paragraphs separating the different subtopics, in particular for NQO1 and radiotherapy.

b)      As copied from ref. [30]: What are "glycolic" metabolic alterations – in biochemical terms?  (you probably do not mean glycolic acid, but glycolytic metabolites.).

c)      Line 147: this sentence is not very informative: what is the message concerning the models?

d)      Briefly explain/define kPA and kP  (Line 151).

e)      Line 164; 166: Instead of using the ackward abbreviations of cell lines (here and elsewhere), please simply state which tumor features they represent. Also, spell out the meaning of PDX.

f)       Ref [28] does not refer to pancreatic tumors, but a glioblastoma model. Please correct.

5. Prostate Cancers

a)      Line 247-8:  what is the reference to this sentence?

b)      Line 253: ref [48] is not correct in this context: It is a publication on renal cancers, which cites studies on prostate cancers. Please correct this sentence and provide a suitable reference.

6. Breast Cancer:

a)      Referring to the publication from Gallagher et al. [15] regarding patients with different breast cancers, it should be mentioned that there was no correlation of malignancy with LDH activities. This is corroborated by studies with breast cancer xenografts (Xu et al. #1) and cell lines in-vitro (Grashei et al. #2). Such a lack of correlation between pyr->lac conversion and malignancy is also reported for hepatocellular carcinoma in ref. [53] (Menzel at al.). Therefore, the general concept of hyperpolarized 13C-pyruvate-to-13C-lactate conversion being correlated with LDH expression and (total) activity as indicators of malignancy is not stringent, which is an important message for the application and interpretation of this methodology.

#1. Xu, H.N.; et al. Is higher lactate an indicator of tumor metastatic risk? A pilot MRS study using hyperpolarized 13C-pyruvate. Acad Radiol 2014, 21, 223-231, doi:10.1016/j.acra.2013.11.014.

#2. Grashei, M.; Conversion of Hyperpolarized [1-13C]Pyruvate in Breast Cancer Cells Depends on Their Malignancy, Metabolic Program and Nutrient Microenvironment. Cancers (Basel) 2022, 14, doi:10.3390/cancers14071845.

b)      Lin 274: "...study revealed that the enzyme lactate dehydrogenase mediated the exchange of the 13C-label..." is a misperception from ref [14], which simply stated that the exchange is catalyzed by this enzyme (which is text book knowledge). Please rephrase this part of the paragraph.

7. Liver Cancer

a)      Line 290: this sentence does not make sense: "detect the impact of cancer on liver function-...

b)      Line 293: this sentence describing the stages and progression of hepatocellular carcinomas needs a reference.

c)      Lines 316-318: Please remove the statistical details (s.d., N-values), as they are not required to highlight the message of this sentence. (This recommendation also pertains to some of the other sections.)

8. Gastric Cancer:

This must be a misspelling: "villain-positive". Please briefly explain the relevance of villin for the metabolic phenotype of this cancer.

9. Melanoma

For the interested reader not "at home" in immunology, please include a brief phrase on what immune checkpoints are in the metabolic context here.

10. All other cancers

Only two (not all other) cancers are addressed in this section: It would make sense to also give renal cancers and leukemia each their own heading.

11. Future directions

a)      Based on the present examples of hyperpolarized 13C-pyruvate-derived imaging: what are its limitations and the future developments for this technology in the clinic. This should be addressed.

b)      In line 424f, please briefly translate these abbreviations and spell out what are the differences between the different imaging techniques: what can (or cannot) they detect? Which advantages would be expected?

c)      How can HP 13C-pyruvate image the microbiome? Reference?

d)      Concerning the statements on bringing together AI and MRSI, ref. [22] (Park et al.) is not a correct choice. Artificial intelligence being a keyword, more information would be expected here – with more appropriate references.

e)     A prerequisite for interpreting the results of HP 13C-pyruvate conversions is our understanding of the potential metabolic pathways and the role of the tumor's microenvironment. Obtaining this in combination with AI deserves a note and reference.

12. Conclusions

a)      This section needs to be rewritten. It should highlight the type of information which has been gained, and what has been learned about the applications with different tumors. Which metabolic changes may be the most informative for the oncologist?

b)      The challenges mentioned here (line 448f.) belong to the section Future Directions.

Author Response

We thank reviewer for his/her insightful comments to enhance the quality of review article. We have addressed the reviewer comments and revised the manuscript as suggested. 

1) In the Title: "Revolutionizing" ... is an overstatement (not suitable in a scientific paper). Surely, the authors have a more appropriate word. (suggestion: New Potentials)

Ans: As suggested changed the title to “Enhancing Cancer Diagnosis with Real-Time Feedback: Tumor Metabolism Through Hyperpolarized 1-13C Pyruvate MRSI”

2) When describing the contents of referenced publications, do avoid naming first authors, as this does not add any relevant information and disturbs the sentence flow and context.

Ans: Suggestion is well received, and the sentences have been modified.

3)  Also, check for redundancies within a sentence (examples: lines 176/179; 331/2; 372/378)

Ans: redundancies in the sentences are corrected.

a)  One of the studies tested a hypoxia-activated prodrug (TH-302) as a monotherapy and in combination with pyruvate (not the HP probe) on three sub-cutaneous (Hs766t, MIAPaCa-2, and SU.86.86 cells) PDX PDAC models derived from different patients [34].

b)  The mouse model didn’t show significant differences on day 0 and day 4 in PET uptake in afatnib treated groups and as well in vehicle treated groups.

c)  One of the studies showed reduction in pyruvate-to-lactate conversion in BRAF mutant melanoma cell line compared to vehicle control. Whereas, increased pyruvate-to-lactate conversion in BRAF wild type melanoma cell line compared to vehicle control[67]. In another study, the combined treatment of BRAF/MEK inhibitors vemurafenib and trametinib in melanoma xenografts showed no significant changes in lactate-to-pyruvate ratios before treatment and after 24 hours of combination treatment.

4) Some paragraphs read more like a collection of reports and lack a narrative. Check for a main thought and a concluding message in each paragraph.

Ans: Corrections have been made as suggested by the reviewer.

5) Avoid abbreviations where possible, as some are very specific to a field in cancer research and not generally familiar. Instead, use a short word /or explanatory term to highlight the relevance of a tumor type (model). However, the abbreviation LDH for lactate dehydrogenase is of common use in cancer-related terminology.

Ans: Suggestions are incorporated in the text.

6) Figure 1 needs revision/amendments: The legend describes hyperpolarized [1-13C]pyruvate. This should also be depicted in the figure. The potential conversion products should also be 13C-labeled and would explain the appearance of 13CO2 from entry of 13C-pyruvate-derived acetyl-CoA into the oxidative TCA cycle.  Also, the path of carboxylation from [1-13C] pyruvate would result in 13C-oxaloacetate and eventually 13C-malate. PC needs to be spelled out.

Ans: Figure 1 is modified as suggested to give a clear picture of the biochemical reaction of [1-13C]pyruvate, and the modified figure can be found in revised manuscript.

7) Figures 2 and 3 – They should be omitted, as they are already published as presented, are too detailed here, and need more explanations for interpretation, which would be out of the scope of this overview. More important here would be to highlight the relevant results and conclusions. Readers interested in the details could consult the referenced publication.

Ans: The figure 2 and 3 are incorporated to showcase representative important outcome of preclinical and clinical studies pictorially to make more interesting to the readers. I agree, readers can always refer to the original paper if they want to go deeper into the study.

Brain cancer

8) Paragraphs 1 and 2 need better phrasing of the lines of thought, i.e. intentions and the message. In line 91: what are the "unique characteristics" relevant to the context?

Ans: Paragraphs have been rephrased and made more specific. “Brain tumors are highly resistant to standard and innovative treatments due to the heterogeneity, hypoxia, and blood brain barrier.”

“HP [1-13C] pyruvate MRSI has shown the potential to distinguish brain tumors in eight human subjects from normal healthy brain based on metabolism.”

9) Line 111: define what is meant by "pseudoprogression"

Ans: As suggested, the definition of pseudoprogression is incorporated in the parenthesis (increase in tumor size is might be due to infiltration of immune cells and accumulation of fluids and followed by the regression of tumor size)

Pancreatic Cancer

10) This section needs a better structure, i.e. more paragraphs separating the different subtopics, in particular for NQO1 and radiotherapy.

Ans: NQO1 and radiotherapy sections are now put in separate paragraphs.

11) As copied from ref. [30]: What are "glycolic" metabolic alterations – in biochemical terms?  (you probably do not mean glycolic acid, but glycolytic metabolites.).

Ans: Thank you for catching this typo- it has been corrected as glycolytic.

12) Line 147: this sentence is not very informative: what is the message concerning the models?

Ans: Line 147 corrected as- “The alanine/lactate ratio was discovered as an imaging biomarker that was decreasing with disease progression from normal tissue to low-grade PanIN to high-grade PanIN and finally to PDAC”

13) Briefly explain/define kPA and kPL (Line 151).

      Ans: These are the kinetic rate constants for Pyruvate to Alanine and Pyruvate to Lactate biochemical reactions. We have now explained this in the text.

14) Line 164; 166: Instead of using the awkward abbreviations of cell lines (here and elsewhere), please simply state which tumor features they represent. Also, spell out the meaning of PDX.

Ans: The cell lines are referred by their standard biological designations. Hs766t line was isolated from a pancreatic carcinoma metastatic to a lymph node. On the contrary, MIA PaCa-2 is an epithelial cell line that was derived from tumor tissue of the pancreas of a male patient and SU.86.86 is a cell line exhibiting epithelial-like morphology that was isolated from the pancreas of female patient with ductal carcinoma. The PDX has been spelled out as patient derived xenografts in line 140.

15) Ref [28] does not refer to pancreatic tumors, but a glioblastoma model. Please correct.

Ans: Thank you. Reference has been corrected to [34] on [Current Line 174]

Prostate Cancers

17) Line 247-8:  what is the reference to this sentence?

Ans: The reference for the Line 247-8 is [46]

18) Line 253: ref [48] is not correct in this context: It is a publication on renal cancers, which cites studies on prostate cancers. Please correct this sentence and provide a suitable reference.

Ans: The sentence with a reference has been moved to the section of renal cell carcinoma.

  1. Breast Cancer:

19) Referring to the publication from Gallagher et al. [15] regarding patients with different breast cancers, it should be mentioned that there was no correlation of malignancy with LDH activities. This is corroborated by studies with breast cancer xenografts (Xu et al. #1) and cell lines in vitro (Grashei et al. #2). Such a lack of correlation between pyr->lac conversion and malignancy is also reported for hepatocellular carcinoma in ref. [53] (Menzel at al.). Therefore, the general concept of hyperpolarized 13C-pyruvate-to-13C-lactate conversion being correlated with LDH expression and (total) activity as indicators of malignancy is not stringent, which is an important message for the application and interpretation of this methodology.
#1. Xu, H.N.; et al. Is higher lactate an indicator of tumor metastatic risk? A pilot MRS study using hyperpolarized 13C-pyruvate. Acad Radiol 2014, 21, 223-231, doi:10.1016/j.acra.2013.11.014.

#2. Grashei, M.; Conversion of Hyperpolarized [1-13C]Pyruvate in Breast Cancer Cells Depends on Their Malignancy, Metabolic Program and Nutrient Microenvironment. Cancers (Basel) 2022, 14, doi:10.3390/cancers14071845.

Ans: We thank the reviewer for the insightful comment. We have added a paragraph describing this nuance in the revised manuscript under the “breast cancer” section and added the references.

 In this paper, lack of correlation of malignancy with LDH activities has been reported. This is corroborated by studies with breast cancer xenografts in vivo [51]  and cell lines in vitro[52].  Such a lack of correlation between pyruvate-to-lactate conversion and malig-nancy is also reported for hepatocellular carcinoma[53]. In fact, emerging quantitative studies have now demonstrated that hyperpolarized [1-13C]pyruvate MRSI measures pri-marily MCT1-mediated [1-13C]pyruvate transmembrane influx in vivo, not glycolytic flux or LDHA activity, driving a reinterpretation of this technology during clinical translation[54, 55]. Therefore, the general concept of hyperpolarized pyruvate-to-lactate conversion being correlated with LDH expression and (total) activity as indicators of ma-lignancy is not stringent and this is an important message for the application and inter-pretation of this methodology.

20) Line 274: "...study revealed that the enzyme lactate dehydrogenase mediated the exchange of the 13C-label..." is a misperception from ref [14], which simply stated that the exchange is catalyzed by this enzyme (which is text book knowledge). Please rephrase this part of the paragraph.

Ans: The paragraph has been modified.

  1. Liver Cancer

21) Line 290: this sentence does not make sense: "detect the impact of cancer on liver function-...

Ans: Sentence has been rewritten as – “One way to detect the cancer in liver is using HP [1-13C] pyruvate MRSI”.

22) Line 293: this sentence describing the stages and progression of hepatocellular carcinomas needs a reference.

Ans: References has been added [References : 56) Chou CT et al. The utility of conventional mr imaging." World J Gastroenterol 19 (2013): 7433-9. 10.3748/wjg.v19.i42.7433   and 57) Kudo M et al. Correlation of imaging with pathology." Journal of gastroenterology 44 (2009): 112-18.]

23) Lines 316-318: Please remove the statistical details (s.d., N-values), as they are not required to highlight the message of this sentence. (This recommendation also pertains to some of the other sections.)

Ans: N values has been removed as suggested and standard deviations are kept as other reviewerS and readers like to see them without referring to the original work.

  1. Gastric Cancer:

24) This must be a misspelling: "villain-positive". Please briefly explain the relevance of villin for the metabolic phenotype of this cancer.

Ans: Typos has been corrected to villin. The relevance was from which cell type the gastric cancer was originated with overexpression of PPARD and metabolic changes in these tumors already has been discussed.

  1. Melanoma

25) For the interested reader not "at home" in immunology, please include a brief phrase on what immune checkpoints are in the metabolic context here.

Ans: The sentence has been added at the beginning of the paragraph (Line 371)- “Immune check point blockade therapy works by blocking the proteins which negatively control the T-cell action. The import proteins of target in ICB are Cytotoxic T-Lymphocyte Antigen-4 (CTLA-4), Programmed Death-1 (PD-1), and Programmed Death Ligand -1 (PDL-1)”

  1. All other cancers

26) Only two (not all other) cancers are addressed in this section: It would make sense to also give renal cancers and leukemia each their own heading.

Ans: Separated as 10. Renal cell carcinoma and 11. Leukemia sections.

  1. Future directions

27) Based on the present examples of hyperpolarized 13C-pyruvate-derived imaging: what are its limitations and the future developments for this technology in the clinic. This should be addressed.

Ans: paragraph on challenges in translation of this technology has been added. Another important challenge is the consistency in obtaining hyperpolarized signals across experiments, which can affect the accuracy and reproducibility of the results. Addressing these challenges through continued research and development and multi-center clinical trials are crucial for the full realization of the potential of this technology in clinical practice

28) How can HP 13C-pyruvate image the microbiome? Reference?

Ans: There are many laboratories that are now working on employing HP to classify different types of microbiome. Two such references has now been added that shows the potential of hyperpolarized metabolic imaging in this exciting application.

29) Concerning the statements on bringing together AI and MRSI, ref. [22] (Park et al.) is not a correct choice. Artificial intelligence being a keyword, more information would be expected here – with more appropriate references.

Ans: This has been corrected with the right reference- Enriquez, J. S., et al. "Hyperpolarized magnetic resonance and artificial intelligence: Frontiers of imaging in pancreatic cancer." JMIR Med Inform 9 (2021): e26601. 10.2196/26601.

30) A prerequisite for interpreting the results of HP 13C-pyruvate conversions is our understanding of the potential metabolic pathways and the role of the tumor's microenvironment. Obtaining this in combination with AI deserves a note and reference.

Ans: We are not aware of any AI-based application of interpreting HP 13C-pyruvate conversions in terms of tumor microenvironment (TME). However, a recent review by Mu et al (https://link.springer.com/article/10.1007/s11307-020-01570-0) has detailed how HP-MRS can interrogate TME  and this reference has been added.

  1. Conclusions

31) This section needs to be rewritten. It should highlight the type of information which has been gained, and what has been learned about the applications with different tumors. Which metabolic changes may be the most informative for the oncologist?

Ans: Rewritten as suggested.

32) The challenges mentioned here (line 448f.) belong to the section Future Directions.

Ans: The paragraph on challenges in technology has been moved to future directions

Round 2

Reviewer 2 Report

The authors have addressed the review comments effectively, and the summary table has been well-prepared in conjunction with Figure 1.